# Who Guides Vaccination in the Portuguese Press? An Analysis of Information Sources

**DOI:** 10.3390/ijerph18042189

**Published:** 2021-02-23

**Authors:** Andrea Langbecker, Daniel Catalan-Matamoros

**Affiliations:** 1Institute of Collective Health, Federal University of Bahia, Salvador 40.110-040, Brazil; 2Research Group for Communication, Health and Education, Department of Public Health, São Paulo State University, Botucatu 18.618-686, Brazil; 3Department of Communication and Media Studies, Madrid University Carlos III, 28903 Madrid, Spain; dacatala@hum.uc3m.es; 4Health Research Centre, University of Almeria, 04120 Almeria, Spain

**Keywords:** content analysis, media, newspaper, public health, sources, journalism, vaccine

## Abstract

Sources of information are a key part of the news process as it guides certain topics, influencing the media agenda. The goal of this study is to examine the most frequent voices on vaccines in the Portuguese press. A total of 300 news items were analysed via content analysis using as sources two newspapers from 2012 to 2017. Of all the articles, 97.7% included a source (*n* = 670). The most frequent were “governmental organisations”, “professional associations” and the “media”. Less frequent sources were “university scientists”, “governmental scientific bodies”, “consumer groups”, “doctors”, “scientific companies”, “NGOs” and “scientific journals”. Most articles used only non-scientific sources (*n* = 156). A total of 94 articles used both categories and 43 used exclusively scientific sources. Our findings support the assertion that media can be an instrument to disseminate information on vaccines. Nevertheless, despite being present in most articles, the number of sources per article was low, therefore not presenting a diversity of opinions and there was a lack of scientific voices, thus suggesting lower quality of the information being offered to the audience.

## 1. Introduction

The media have shown increasing interest in the dissemination of health content as a reflection of how important this topic is in our contemporary society [1,2]. As architects of the social imaginary, the media can play a key role in the dissemination of quality health-related information [3]. This is crucial if we believe that information and ideas provided by journalism can have effects and influence individual or collective decisions on health [4].

Many studies on journalism have focused on understanding the processes related to information production, which is the result of complex social interaction amongst social players, amongst them journalists and their information sources [5,6,7,8]. The sources are people (speaking on their behalf or as members or representatives of an institution or organisation) who offer information or suggestions to journalists when drafting a news item [5,6,9]. But sources are also any material coming from other media, news agencies, scientific journals, amongst others [10]. An indicator of journalistic quality is offering the audience varied content with a wide range of viewpoints and sources [11].

From the point of view of the agenda setting theory, sources can play a key role when creating public opinion on specific content, as well as influencing the approach of a news piece (framing) [10,12]. This point of view is even more relevant when we refer to the press coverage of issues related to public health because of their high social impact, as is the case with vaccines. Vaccination has improved the public health scenario since the 20th century, reducing the high mortality rates of the past. They were seen as a great step for humankind and as a basic health intervention and a priority worldwide [13,14]. 

Despite those benefits [15], there are decreases in vaccination rates globally, mainly in the United States and in some European countries, with an increase on the outbreaks of diseases that can be easily prevented with vaccination [16,17,18]. This situation can be due, amongst other factors, to anti-vaccine movements present in the public space and that can discredit the security and benefits of vaccines [19]. The analysis of the media can be an important tool to understand their role in this context.

Nevertheless, there are few studies that focus on information sources and health content [10,20,21,22], and those related to information sources and vaccines are even fewer [23,24], highlighting a clear gap in this line of research in the field of health news. In this sense, it is appropriate to study this topic, as it plays a key role in news production, being an indicator of journalistic quality. The goal of this study is to examine information sources on vaccines in the Portuguese press, thus identifying those who guided the topics and discussions on vaccination, presenting a general view of press coverage on that topic.

### Information Sources

The use of sources is the object of research and theoretical discussion in journalism studies, increasingly more so since the seventies [6,7,8,9,25,26,27,28]. The media have an important influence on the way people perceive problems [29], and journalists tend to frame their coverage based on the sources they use [30]. McCombs and Shaw [31] already questioned whether journalists were the ones establishing the agenda or whether it was just a reflection of the agenda of their information sources.

Journalists move within a network made up of a strategic organisational mechanism on which information sources move quite effectively to ensure the constant flow of reliable news items [8]. Several criteria are used to assess the quality of information, such as the authorship, productivity and credibility [5]. Sometimes, journalists use a source because of what it is rather than for what information it provides. The more prestigious the title or position of an individual, the greater the confidence on her authority. This is known as credibility hierarchy [5,8]. Gans [6] added that one of the most crucial factors when choosing a source is its ability it provide suitable information. Social and geographical closeness between journalists and sources also have an influence on that process. 

The relationship between the media and their sources is still one of the most complex issues of the whole news-production process [9], with constant inter-relationships and disputes [27], a relationship also marked by mutual interest [9]. As Bourdieu [32] stated, there are no selfless acts. Journalists want information and there are sources interested in providing it. They stop being passive sources and, in many cases, there are newsworthiness acts whose goal is to capture the attention of the media [33].

## 2. Materials and Methods

The article hereby follows a quantitative methodology via content analysis, a set of communication analysis techniques, using systematic procedures through categorization. Such a method allows the classification of the elements of meaning encompassed by the message [34]. Portuguese mainstream newspapers *Diário de Notícias* and *Jornal de Notícias* were analysed. *Diário de Notícias* has its headquarters in Lisbon and was founded in 1864. It is the oldest running newspaper in the country and it is considered to be a reference newspaper with a close relationship with the Portuguese society [35]. Founded in 1888 and with a more popular profile [36], *Jornal de Notícias* is headquartered in the city of Porto, in the north of Portugal.

The period studied was from 1 January 2012 till 12 December 2017. Searches were carried out through the international news database Factiva. Portuguese keywords were used [vacin* OR imuniza*], and they had to be present in the headline or subheading. All news pieces published during that period (news items, short reports, interviews, stories or opinion pieces) were included in the search. Duplicated articles were excluded from the sample, as well as those that used the term “vaccine” with a metaphorical meaning. 

Articles were read and re-read in order to classify the types of vaccines and most frequent frames used in the pieces analysed. Frames were classified into; conflict, human interest, financial consequences, morality and responsibility allocation [37]. When the same article could be related to more than one frame, it was classified under the most prevailing one. The prevailing frame was considered to be the one that encompassed the entire article (including titles and body of the text), thus being more representative of the predominant approach in the text. In 7% of the articles, it has been identified more than one frame. Regarding information sources, they were classified as: governmental organisations (Ministry of Health and its regional health administrations and international organisations), professional associations (including any association made up of health professionals and their members such as the Portuguese Society of Paediatricians), media (news agencies, newspapers, TV channels), university scientists (scientists linked to universities or research centres), governmental scientific organisations (such as the National Institute of Health Doctor Ricardo Jorge), consumer groups (patients, patient representatives, user associations’ representatives), NGOs (Non-governmental organisations), doctors (including any healthcare professional), scientific companies (including the pharmaceutical and technological sectors), scientific journals (scientific peer-review publications). The category “others” was used for all sources that did not fit any of the above categories.

For coding reliability, one researcher (AL) performed a first coding round, while a second round was performed by another researcher (DCM). Any discrepancy was resolved with the support of a third researcher (CPS) when necessary in order to reach 100% of agreement.

## 3. Results

Between 2012 and 2017 a total of 337 information pieces on vaccines were published in the newspapers analysed. A total of 37 articles were excluded (Figure 1).

The final sample included 300 articles. Of those, *Jornal de Notícias* published 166 articles and *Diário de Notícias* 134 texts, without significant differences amongst them (*χ*^2^ = 3413; *p* = 0.065; df = 1). The most common were texts related to the flu vaccine (*n* = 87, 29.8%), measles (*n* = 33, 11.3%) and meningitis (*n* = 28, 9.6%), accounting for over 50% of the articles (Table 1). The most frequent frames were “human interest” (*n* = 170, 56.67%) and “responsibility” (*n* = 83, 27.67%) (Table 1).

The human interest group included articles related to how vaccines could help individuals or specific groups, with statements such as: “*The vaccine against cervical cancer is also beneficial for men*” [38]. Responsibility was present both in the sense of making health authorities responsible for something that had happened but also to recognise it as a solution to a problem or difficulty: “*Illegal import of vaccines threatens national security*” [39]. Other frames were found, such as the economic, conflict and morality frames, but they were less common.

Most of the articles presented some source of information (293, 97.7), as shown in Table 2. A total of 670 sources were used in the 300 articles published, with an average of 2.23% sources per article. There were seven articles that did not mention any source (2.3%). One source was used by 110 articles (36.6), 86 articles used two sources (28.6) and 46 used three sources (15.3). All those articles accounted for 80.5% of the texts. From five sources onwards the number of articles started decreasing drastically, and there were 11 articles that used five sources and five articles that used from seven to 11 sources. 

In terms of classification, “governmental organisations” were the most frequent source (52.3, *n* = 351), followed by “professional associations” (10, *n* = 67) and “media” (7.1, *n* = 48). Other sources found to a lesser extent were “university scientists”, “governmental scientific bodies”, “consumer groups”, “doctors”, “scientific companies”, “NGOs” and “scientific journals”. 

Furthermore, sources were classified into two large groups: scientific and non-scientific. The former included the following categories: “governmental scientific organisations”, “professional associations”, “university scientists”, “doctors” and “scientific companies”. The non-scientific included: “governmental organisations,” “NGOs”, “media”, “consumer groups” and “others”. 

Most articles used only non-scientific sources (*n* = 156). Contrary to this, 94 texts used both categories and 43 texts were based solely on scientific sources. 

In the non-scientific sources, one of the most frequently category was “governmental organisations” with the prevalence of source as General Directorate for Health with 39.9, followed by Ministry of Health (8.54), World Health Organization (OMS) (7.4), National Medicines and Health Products Authority (Infarmed, for its acronym in Portuguese) (5.4) and Regional Health Administrations (4.8). Other sources were varied and less frequent in this category, without an expressive concentration.

Although we have analysed a health topic, scientific sources were less represented, with a presence in 30.44% of the sample. We observed the repetitive use of the same specialised source throughout the press coverage. For instance, in “governmental scientific associations”, there was massive use of the National Institute of Health Doctor Ricardo Jorge (51.4). In “professional associations”, there was the prevalence of the Portuguese Society of Pulmonology (26.9) and the Portuguese Society of Paediatrics (20.9). As another example, in the “doctors” category, the pulmonologist interviewed was always the same one, as was the paediatrician used as a specialised source.

We also carried out a comparison between scientific and non-scientific sources with the frames and journalistic genres (Table 3). Texts that did not use sources were short items and used a human interest frame. Both news pieces and short items preferred non-scientific sources. News stories mostly used both categories. Non-scientific sources were also the most widely used in the most frequent frames: human interest and responsibility. 

## 4. Discussion

This article studies Portuguese press coverage on vaccines, focusing on information sources. To achieve this goal, 300 articles related to the topic were analysed from two national newspapers, from 2012 to 2017. Most articles used some kind of source. Despite that, there is a large number of articles that used up to three sources per article, suggesting a loss of information quality in this analysed sample. In order to avoid having journalistic content that is merely a reproduction of a few sources, it needs to offer a diversity of voices and perspectives [9].

Regarding their classification, governmental organisations were the most frequent source, highlighting their influence in the construction of the media’s agenda on topics related to vaccination. Official sources accounted for a cohesive and uniform block; they are the ones considered trustable. One way the media used to ensure reliability was to present institutional voices that are socially acknowledged as the most relevant, leaving aside their importance or authority [5,40]. This study confirms the trend to give greater importance to official sources over other sources. This was also observed in previous studies [10,23,41]. An example was the case of General Directorate for Health, responsible for coordinating disease prevention and health promotion activities in Portugal, thus suggesting its highly influential role as a source in the press coverage on vaccines.

Expert sources give greater credibility to news items when compared to sources and are not related to health [42], proof of how important it is for the media to resort to those sources to provide their audience with more credible and scientific-backed content. Specialised sources play a key role in the dissemination of knowledge, having a positive influence in how people see vaccines and their benefits. They are considered to be even more reliable because they offer neutral information, without conflicts of interest [40]. 

It is important to consider that the overwhelming use of official sources somehow replicates the existing power structure in our society, an ideological construct, due to the relationship between those sources and the media and the former’s privileged access as an accredited source [8,40]. However, health news may be specific enough to justify that dependency on official sources or health-related expert sources [41].

Related to repetitive use of the same specialized source, we identified the case of National Institute of Health Doctor Ricardo Jorge linked to the Ministry of Health. Founded in 1899, the Institute is active as a state laboratory in the health sector and a national health observatory. Despite the importance and continued trajectory of the institute in the context of Portuguese public health, we highlight the recurring use of the same source of information. More stable sources respond better to the routine of news production, making the job of the journalist easier by offering clear and brief information, something frequently sought by the journalists, therefore encouraging them to use always the same sources [8,23,43]. 

Despite a predominance of official sources, other social players may try to influence the press, such as the case of citizens as sources, even if this struggle for predominance is mostly unbalanced [43]. Democratic societies seem to view favourably the use of general public as a source by the media [44]. They include in that group patient associations’ representatives who mobilise themselves to gain visibility in the media [5]. They also include in this category people who have witnessed events, providing information or giving their opinion as citizens impacted by the facts, as well as those who contribute giving context to facts and also those who express their anxieties or their own questioning of a specific topic [43]. 

In this study, the category “consumer groups” was hardly represented (5.4%), in agreement with the results of Gomes and Lopes [23] in an analysis on the sources of Portuguese press. Despite how scarcely representative this category is, there was a relevant use of individual citizen sources, with the use, for instance, of stories of those who had been vaccinated against a specific disease or that did not have access to a vaccine. This is a journalistic resource called personification, giving visibility to people involved in a specific event, stressing the human factor of the news piece so that readers can connect more easily to the story [45]. For Hinnant et al. [46], journalists covering health topics resort to personal stories to put a face on an issue related to health, stressing that journalists choose people with the aim of informing and inspiring others on that topic. 

Although the coverage included mostly sources that approached vaccination as a preventive measure, there was also some space for sources against immunisation. News items did not always compare different points of view or presented a specialised source on the topic, as in the example below: “*A mum looked into the issue and decided not to vaccinate*” [47] where the only source was the mother, giving weight to her argument against vaccination. This practice does not follow a premise of ethical journalism: to offer a range of points of view and opinions [48], mainly because the news on health should offer enough tools for people to make the best decisions on their health [41]. 

The quest for a balance between sources is a complex topic in journalism. It is generally considered that journalists should allow for different opinions to have a voice in news items, that is, they should offer both sides of the story [48,49]. Nevertheless, there is no consensus on which conditions can really guarantee that balance, or if different sources should have the same space or weight [48], even more so in items related to scientific content, where a non-scientific source should not be compared to a scientific source [50]. We believe this is an important reflection for health-related news items, as is the case for vaccines, bearing in mind that the circulation of certain content could affect people, making them take wrong decisions on their health. 

In relation to topics that have high social relevance, such as the case of vaccines, it is essential that journalists ensure the reliability of their sources, considering the proliferation of misinformation and fake news related to this subject [24]. As an example, in the current coronavirus pandemic, the predominant speeches in social media, showed increasing circulation of false and misleading information related to Covid-19 [51]. As Catalan-Matamoros and Elias [24] pointed out, one of the ways that journalists should resort to dealing with this situation is to use a range of information sources, in addition to government sources, thus guaranteeing the plurality of voices. By improving the quality of information, it would be possible to achieve better informed and more aware citizens.

For Feemster [52], any effective vaccine policy initiative must be complemented with effective communication, based on reliable and accurate information that gives guidance to people in making decisions about their health.

This study has some limitations. We focused on Portuguese written press and the search did not include all media, such as TV, radio or news websites. Considering that each medium has its specificities regarding the selection of information sources, these findings may not apply to all. Stroobant et al. [10], for instance, observed large differences in the use of sources amongst the different media when analysing the Belgian media. It would be relevant for future research to use qualitative techniques, such as interviews with journalists or editors about the selection of sources in order to provide other viewpoints on the topic.

Additionally, this article focuses on one single country and this represents a geographic limitation about the results. However, recent systematic reviews [53,54] have found a lack of research about public communications on the vaccine in non-English speaking countries. Therefore, our study fills this gap in current research about vaccines.

## 5. Conclusions

Information sources have brought the topic of vaccines to the agenda of the Portuguese newspapers studied. The findings of this study support the belief that the media can be an instrument to disseminate information on vaccines. However, although information sources appeared in most articles, there was low diversity of sources, therefore suggesting a loss of the quality of information offered to their audience. The most common sources were governmental, professional organisations and the media. Nevertheless, even though the topic was related to health issues, significantly less space was given to scientific sources. Citizen sources were a less-used resource, reinforcing the idea that press coverage gives more space to institutional sources. We hope this study improves and enhances our knowledge on the role of sources in press coverage on vaccines. Anti-vaccination movements are spreading in many countries, therefore it is necessary to delve into the voices driving the vaccination discourse in the media.

## Figures and Tables

**Figure 1 ijerph-18-02189-f001:**
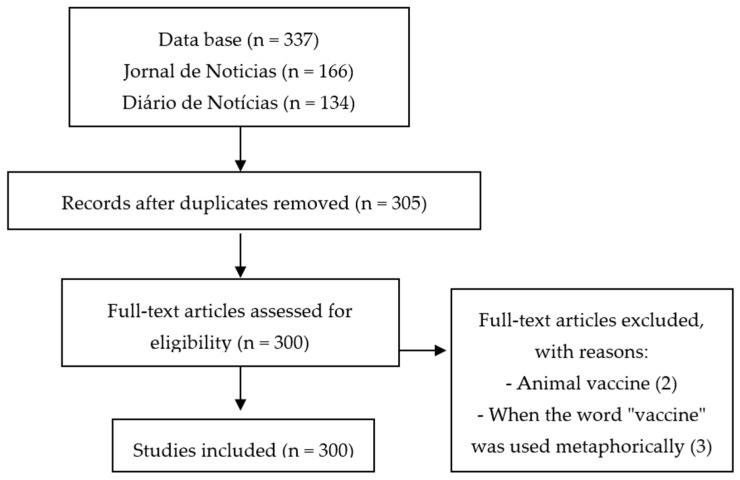
Search strategy and synthesis of the process of obtaining the selected articles.

**Table 1 ijerph-18-02189-t001:** Characteristics of the study sample.

**Journalistic Genres**	**N**	**%**
News	208	69.3
Feature	61	20.3
Short news	18	6
Opinion	8	2.6
Interview	5	1.6
Total	300	100.0
**Type of Vaccine**	**N**	**%**
Influenza	88	29.3
Measles	37	12.3
Meningitis	28	9.3
Tuberculosis	21	7
Tetravalent/Pentavalent	15	5
Hepatitis	12	4
Meningitis/rotavirus	10	3.3
Ebola; Human Papillomavirus	9	3
HIV	6	2
Malaria	5	1.6
Polio	4	1.3
Alzheimer disease; Cancer; Zika	3	1
Leprosy; Cutaneous leishmaniasis	2	0.6
Intestinal bacteria; Dengue; Yellow fever; Tetanus/Diphtheria; Nasal vaccine Respiratory virus	1	0.3
General/No identified	39	13
Total	300	100.0
**Frames**	**N**	**%**
Human interest	170	56.63
Responsibility	83	27.6
Economic	25	8.3
Conflict	20	6.6
Morality	2	0.6
Total	300	100.0

**Table 2 ijerph-18-02189-t002:** Frequency counts for sources.

**Type of Sources**	**N**	**%**
Government organizations	351	52.3
Professional associations	67	10
Media	48	7.1
University scientists	40	5.9
Government scientific organizations	35	5.2
Consumer groups	35	5.2
Clinicians	28	4.1
Scientific companies	18	2.6
NGOs	18	2.6
Scientific journals	16	2.3
Others	14	2.08
Total	670	100.0
**Category of Sources**	**N**	**%**
Non-Scientific sources	466	69.55
scientific sources	204	30.44
Total	670	100.0
**Number of Sources Per Article**	**N**	**%**
0	7	2.3
1	110	36.6
2	86	28.6
3	46	15.3
4	30	10
5	11	3.6
6	5	1.6
7	1	0.3
8	1	0.3
9	1	0.3
10	1	0.3
11	1	0.3
Total	300	100.0

**Table 3 ijerph-18-02189-t003:** Frequency of articles by sources, journalistic genre and frames.

**Journalistic Genre**	**Non-Scientific Sources**	**Scientific Sources**	**Scientific Sources and Non-Scientific**	**No Source**
News	111	27	70	-
Shorts News	36	12	6	7
Feature	2	0	16	-
Opinion	4	2	2	-
Interview	3	2	0	-
Total	156	43	94	7
**Frames**	**Non-Scientific Sources**	**Scientific Sources**	**Scientific Sources and Non-Scientific**	**No Source**
Human interest	78	33	52	7
Responsibility	48	6	29	-
Economic	19	1	5	-
Conflict	9	3	8	-
Morality	2	0	0	-
Total	156	43	94	7

## Data Availability

Not applicable.

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
