# Peer review of "Who Guides Vaccination in the Portuguese Press? An Analysis of Information Sources"

_ijerph, 2021, doi:10.3390/ijerph18042189_

Round 1

Reviewer 1 Report

1. It would be interesting to note whether there were also changes in the population's views on vaccinations based on the articles presented through the media. Have vaccination rates been increasing over the given period or have they changed because of the media presentation?

2. It would be interesting to note whether the sources used by the media accurately portraying vaccinations and the need for vaccinations to the population or are they using popular sources that lack discrete data to base conclusions.

3. It would be of interest to mention in the discussion how the media is approaching the vaccinations surrounding the coronavirus. Is the media using the same lack of appropriate sourcing in the mentions of the need for the coronavirus vaccine? 

Author Response

Authors’ comments:  Thank you for your feedback. We sincerely appreciate your review, which has undoubtedly helped to improve the manuscript. Please find our answers to your specific comments below.

  1. It would be interesting to note whether there were also changes in the population's views on vaccinations based on the articles presented through the media. Have vaccination rates been increasing over the given period or have they changed because of the media presentation?

Authors’ comments: Thank you for pointing this out. Although these two perspectives are quite relevant, they fall out of the scope of our research, which aimed to analyze the sources of information in journalistic coverage related to vaccines. If there were changes in people's vision towards vaccination, it would be necessary to develop a study about the public's perception of the news. Concerning the comparison of journalist coverage with vaccination rates, we have already carried out a previous study regarding such this interesting perspective:

Daniel Catalan-Matamoros & Carmen Peñafiel-Saiz (2020) Exploring the relationship between newspaper coverage of vaccines and childhood vaccination rates in Spain, Human Vaccines & Immunotherapeutics, 16:5, 1055-1061, DOI: 10.1080/21645515.2019.1708163

  1. It would be interesting to note whether the sources used by the media accurately portraying vaccinations and the need for vaccinations to the population or are they using popular sources that lack discrete data to base conclusions.

Authors’ comments: This possibility is also quite interesting. Our data suggest a lack in the quality of the information because most of the articles did not use scientific sources. However, we have not included the content quality variable in our analysis. We will take it into account in future researches.

  1. It would be of interest to mention in the discussion how the media is approaching the vaccinations surrounding the coronavirus. Is the media using the same lack of appropriate sourcing in the mentions of the need for the coronavirus vaccine? 

Authors’ comments: Following the reviewer´s recommendation, we have included a paragraph related to coronavirus pandemic in the discussion.

Reviewer 2 Report

The manuscript "Who guides vaccination in the Portuguese press? An analysis of information sources" presents a quantitative content analysis of two major portuguese newspapers in the period 2012-2017 about the sources used in articles mentioning vaccines. The topic of the manuscript is highly interesting but several aspects of the presentation need to be improved, especially in order to insure the results can be replicated.

Major Observations

1) In line 84 authors mention "quantitative methodology via content analysis [34]." The manuscript will benefit from a concise and informative introduction to what content analysis is and its associated quantitative methods are either in the methods or introduction section of the manuscript.

2)In lines 101-103 authors mention "When the same article could be related to more than one frame, it was classified under the most prevailing one." It is imperative that authors clearly specify the objective criteria used to consider the most prevailing topic. For multiple sources the information is shown in table 2 and i wonder if the number of sources is correlated with the number of "frames".

2.1)Alternatively, it will be highly informative if authors present the data on how many articles present dual data.

3) Also, please present the data about the disagreements between the two researchers scoring the articles referred in lines 115-117.

4)Line 123, what was the specific hypothesis tested with chisquare? What proportion or number of keywords or what was being tested?

5) Table 1 will be more informative if data are disaggregated by journal. That way is easier to compare similarities between the two newspapers. 

6) Lines 138-144, for the super common sources, please identify them here in the results, and then elaborate more about their role in the context of portuguese society. For example, in the results you can directly mention the "National Institute of Health Doctor Ricardo Jorge" and then explain why it is a key source for information about vaccines in the discussion, for example, stating the mission of such institute.

7) It will be best to present how the inclusion exclusion criteria for the articles were implemented using a flow diagram like the ones used in papers describing meta-analyses in the biomedical literature. Please check, for example, the following document for further guidance:

https://pubmed.ncbi.nlm.nih.gov/10789670/

8) The ideas in the discussion make sense in light of the data. However, the discussion and all sections in the manuscript require a dedicated proofreading and english edition, since currently the language employed through the manuscript has many issues related to a poor quality english. I am attaching a few comments i made as i read and evaluate the manuscript (please note they are not exhaustive)

Author Response

The manuscript "Who guides vaccination in the Portuguese press? An analysis of information sources" presents a quantitative content analysis of two major portuguese newspapers in the period 2012-2017 about the sources used in articles mentioning vaccines. The topic of the manuscript is highly interesting but several aspects of the presentation need to be improved, especially in order to insure the results can be replicated.

Authors’ comments:  Thank you for your positive feedback. We appreciate your review very much which has undoubtedly helped to improve the manuscript. Please find our answers to your specific comments below.

In line 84 authors mention "quantitative methodology via content analysis [34]." The manuscript will benefit from a concise and informative introduction to what content analysis is and its associated quantitative methods are either in the methods or introduction section of the manuscript.

Authors’ comments: Following the reviewer´s recommendation, we added the concept of content analysis in the methods section.

2)In lines 101-103 authors mention "When the same article could be related to more than one frame, it was classified under the most prevailing one." It is imperative that authors clearly specify the objective criteria used to consider the most prevailing topic. For multiple sources the information is shown in table 2 and i wonder if the number of sources is correlated with the number of "frames".

2.1)Alternatively, it will be highly informative if authors present the data on how many articles present dual data.

Authors’ comments: The frame category was used in the article to present further elements that characterize journalist coverage in Portugal, although it is not the main focus of the investigation. Following the reviewer's recommendations, we have added this information to the text.

3) Also, please present the data about the disagreements between the two researchers scoring the articles referred in lines 115-117.

Authors’ comments:  Following the reviewer's recommendations, we added a paragraph in the text to clarify this information.

4)Line 123, what was the specific hypothesis tested with chisquare? What proportion or number of keywords or what was being tested?

Authors’ comments:  Chi-square analysis was employed to determine whether the category distribution significantly differed concerning the number of articles, comparing the two journals included in our sample.

5) Table 1 will be more informative if data are disaggregated by journal. That way is easier to compare similarities between the two newspapers. 

Authors’ comments: Thank you very much for your specific feedback. The objective of the study was to present a sample of articles that represent journalistic coverage of vaccines in the Portuguese press. It was not meant to make a comparative study between the two newspapers. We also believe that the scope of the journal is more aligned to this national and general approach taking into account that this is an international journal.

6) Lines 138-144, for the super common sources, please identify them here in the results, and then elaborate more about their role in the context of portuguese society. For example, in the results you can directly mention the "National Institute of Health Doctor Ricardo Jorge" and then explain why it is a key source for information about vaccines in the discussion, for example, stating the mission of such institute.

Authors’ comments: Following the reviewer's recommendations, we added a paragraph related to this subject.

7) It will be best to present how the inclusion exclusion criteria for the articles were implemented using a flow diagram like the ones used in papers describing meta-analyses in the biomedical literature. Please check, for example, the following document for further guidance:

https://pubmed.ncbi.nlm.nih.gov/10789670/

Authors’ comments: Following the reviewer's recommendations, we added a flow diagram in the results.

8) The ideas in the discussion make sense in light of the data. However, the discussion and all sections in the manuscript require a dedicated proofreading and english edition, since currently the language employed through the manuscript has many issues related to a poor quality english. I am attaching a few comments i made as i read and evaluate the manuscript (please note they are not exhaustive).

Authors’ comments:  Thank you for your specific feedback. We have gone through your grammar and proofreading comments in the text. Additionally, and once the manuscript review process is completed, we will submit the entire manuscript to a native professional proofreading company based in the United Kingdom.

Round 2

Reviewer 1 Report

Thank you for your revisions. No additional comments

Reviewer 2 Report

Comments were successfully addressed.  Only remaining issue is a detailed edition of the english language.